# Supporting regional pandemic management by enabling self-service reporting—A case report

Richard Gebler[1]*, Martin Lehmann[1], Maik Löwe[1], Mirko Gruhl[1], Markus Wolfien[1], Miriam Goldammer[1], Franziska Bathelt[1,2], Jens Karschau[3], Andreas Hasselberg[3], Veronika Bierbaum[3], Toni Lange[3], Katja Polotzek[3], Hanns-Christoph Held[4], Michael Albrecht[5], Jochen Schmitt[3‡], Martin Sedlmayr[1‡]

1 Institute for Medical Informatics and Biometry, University Hospital Dresden and Medical Faculty Carl Gustav Carus, TU Dresden, Dresden, Germany, 2 Thiem-Research GmbH at Carl-Thiem-Clinic, Cottbus, Germany, 3 Center for Evidence-Based Healthcare, University Hospital Dresden and Medical Faculty Carl Gustav Carus, TU Dresden, University Hospital Carl Gustav Carus Dresden, Dresden, Germany, 4 Clinic and Polyclinic for Visceral, Thoracic and Vascular Surgery, University Hospital Carl Gustav Carus Dresden, Dresden, Germany, 5 University Hospital Carl Gustav Carus Dresden, Dresden, Germany

☯ These authors contributed equally to this work.
‡ JS and MS also contributed equally to this work.
* richard.gebler@tu-dresden.de

**Data Availability Statement:** While privacy and security concerns prevent the disclosure of specific data relating to pandemic management in local hospitals, we wish to clarify that such data isn't the

## Abstract

### Background

The COVID-19 pandemic revealed a need for better collaboration among research, care, and management in Germany as well as globally. Initially, there was a high demand for broad data collection across Germany, but as the pandemic evolved, localized data became increasingly necessary. Customized dashboards and tools were rapidly developed to provide timely and accurate information. In Saxony, the DISPENSE project was created to predict short-term hospital bed capacity demands, and while it was successful, continuous adjustments and the initial monolithic system architecture of the application made it difficult to customize and scale.

### Methods

To analyze the current state of the DISPENSE tool, we conducted an in-depth analysis of the data processing steps and identified data flows underlying users' metrics and dash-boards. We also conducted a workshop to understand the different views and constraints of specific user groups, and brought together and clustered the information according to content-related service areas to determine functionality-related service groups. Based on this analysis, we developed a concept for the system architecture, modularized the main services by assigning specialized applications and integrated them into the existing system, allowing for self-service reporting and evaluation of the expert groups' needs.

### Results

We analyzed the applications' dataflow and identified specific user groups. The functionalities of the monolithic application were divided into specific service groups for data

central focus of our study. Instead, our attention is primarily concentrated on the creation and practical implementation of a self-service system for managing pandemics. As a testament to our commitment to transparency, reproducibility, and further progression in our research, we have made the entire codebase of our system publicly available. This includes configuration files and an illustrative example of data transfer. To provide a tangible sense of the system's capabilities, we have also included a sample dashboard demonstrating the output. Researchers and other individuals interested in our work can gain access to these resources via our GitHub repository at https://github.com/cDataPusher/SelfServeModularPandemicManagement/tree/main. Our ultimate objective in providing these resources is to empower our peers within the scientific community to reproduce the self-service system, adapt it to their distinct requirements, and contribute to its ongoing refinement. We remain staunch in our belief that open collaboration and knowledge sharing are pivotal forces in driving our collective response to global health emergencies, such as the COVID-19 pandemic, onwards and upwards.

**Funding:** The study was funded by the Saxon Ministry for Social Affairs. The funder had no role in study design, data collection and analysis, decision to publish, or preparation of the manuscript. The Article Processing Charge (APC) was funded by the joint publication funds of the TU Dresden, including Carl Gustav Carus Faculty of Medicine, and the 'Sächsische Landesbibliothek – Staats- und Universitätsbibliothek' (SLUB) Dresden, as well as the Open Access Publication Funding of the DFG.

**Competing interests:** Outside the scope of this study, Jochen Schmitt has received consultation fees from Novartis, Lilly, and Sanofi, and his institution has been granted funding for investigator-initiated research from Novartis, Sanofi, Pfizer, and ALK. Michael Albrecht holds the position of CEO at the University Hospitals Carl Gustav Carus Dresden. We wish to clarify that these affiliations do not influence our commitment to PLOS ONE's policies on data and material sharing. This does not alter our adherence to PLOS ONE policies on sharing data and materials.

processing, data storage, predictions, content visualization, and user management. After composition and implementation, we evaluated the new system architecture against the initial requirements by enabling self-service reporting to the users.

## Discussion

By modularizing the monolithic application and creating a more flexible system, the challenges of rapidly changing requirements, growing need for information, and high administrative efforts were addressed.

## Conclusion

We demonstrated an improved adaptation towards the needs of various user groups, increased efficiency, and reduced burden on administrators, while also enabling self-service functionalities and specialization of single applications on individual service groups.

## 1 Introduction

The COVID-19 pandemic exposed shortcomings in the collaboration among research, care, and management in Germany and worldwide [1–3]. In the early stages of the crisis, there was an urgent need for timely and accurate information to track the spread of infections [4, 5]. In Germany, monitoring of the spread of infections initially focused on the overall situation [6]. To this end, the local health authorities reported the confirmed infections to the responsible federal authority, the Robert-Koch-Institute (RKI), in accordance with the German Infection Protection Act (Infektionsschutzgesetz). The RKI subsequently published the count of these infections on a daily basis in an aggregated form [7].

### 1.1 The rise of regional data management needs

As soon as local outbreaks emerged, many occurred at healthcare facilities like nursing homes, affecting particularly vulnerable groups. This situation heightened the demand for more granular regional information. A primary focus became bed management in hospitals, as there was a need to effectively allocate healthcare resources to areas with high infection rates.

The need for regional management required direct exchange of data between health authorities and local stakeholders. This exchange could not be established rapidly in a structured way, firstly due to differing laws and regulations across federal states, but also due to a low level of digitalization, the lack of data exchange interfaces, interoperability and privacy protection regulations. As a consequence, various custom implementations were established both on the national and regional level. On the national level, dashboards provided by the German Interdisciplinary Association for Intensive Care and Emergency Medicine (DIVI) [8], the University Hospital Bonn in collaboration with the Network University Medicine (NUM) and the Medical Informatics Initiative Germany (Medizininformatik-Initiative, MII) [9], aimed to provide relevant and up-to-date information to support decision-making in combatting the pandemic. These reporting tools were targeted towards specific stakeholders such as clinicians, policy as well as decision makers. However, they did not offer regional information on up-to-date clinical capacity or even forecasts required for optimal resource allocation, e.g. allocation of COVID-19 patients. In the federal state of Saxony, the regional coordination centers located at three hospitals of maximal care of the state communicated the need for a hospital-level

overview of COVID-19 bed capacities for normal and intensive care for all clinics. To meet this need, hospitals were asked to report data on the utilization and capacities of normal and intensive care beds for COVID-19 patients on a daily basis.

These data, combined with reported case counts of infection from health authorities and other publicly available information were made accessible as part of the so-called DISPENSE project ("Dresden Information and Forecasting Tool for Bed Utilization in Saxony"), which provided data for the Prime Minister and formed a major decision-making resource for Corona policy in Saxony since march 2020.

## 1.2 The initial challenges with DISPENSE

The project initially consisted of a dashboard, as well as automated reports updated on a daily basis. It contained infection numbers, information about the current capacities of inpatient care as well as model-based predictions of those capacities on a forecasting horizon up to 2 weeks [10]. The authors of this paper were deeply involved in the development and operational aspects of this project, working alongside other professionals to ensure its effectiveness.

This information was provided for the two main stakeholders of the DISPENSE project, the clinic coordination centers and the Saxon Ministry for Social affairs, as it was in high demand at both the political and the clinical level to enable timely action as part of a regional pandemic management. Given the authors' involvement in this project, which ranged from data collection, preparation, and forecasting to system development and dashboarding, the insights shared here come from their first-hand experience. DISPENSE addressed the unpredictable situation caused by the COVID-19 pandemic and through development of a custom web application as a quick response. It was an open source R Shiny Server application [11] that extracted, transformed, and stored the data from various sources. It then processed and visualized this data in the form of tables and interactive graphs at different levels of detail during runtime. In addition, a detailed PDF report summarizing the information provided by the dashboard was generated on a daily basis. The application had a focus on statistical modeling and allowed for the implementation of automated forecasting of bed capacities based on the dynamics of the pandemic in the past weeks in a short period of time. At the start of the project, a monolithic system architecture like this was pragmatic because the development and deployment were limited to a single application that contained all the logic, functionality, and underlying processes, and could be written in a single programming language [12, 13]. The low system complexity of this stage also facilitated important aspects such as management, monitoring, and versioning of the application, compared to more modular approaches like microservices or service-oriented architectures [14]. Additionally, at the beginning of the development phase, it was not clear what kind of data, data sources, and key figures would become relevant in the future or which user groups would use the software. The adaptability of the platform, i.e. incorporation of rapidly evolving scientific knowledge regarding the pandemic into the dashboard and the reporting system, were a condition posed by the Saxon Ministry for Social Affairs.

The dynamic changes in pandemic management and the corresponding need for information led to frequent adjustments to the application, including:

- Development and implementation of new metrics, including retrospective views (e.g., hospitalization incidence, occupancy incidence), as well as monitoring of the effects of pandemic response measures,

- Changes to data sources and comparison of data across other sources, including harmonization processes, which led to data loss and application failures,

- Creation of different visualizations of the data for different user groups, including adapting the layout for expert panels or presentations and

- Integration of new user groups with new requirements for different functionalities, such as the Microbiology for gene sequencing or practicing physicians (e.g., search function, sorting functions, filter functions, notification when certain events occur or threshold values are reached).

Due to its initial monolithic system architecture, the application quickly became more complex and difficult to customize, with very limited scalability due to the dependency of all of its components. Additionally, new user groups expressed their wish to develop and provide their own analyses and evaluations on subject-specific data sets. However, the complexity of the application made it difficult to guarantee independent extensibility of its functionalities and representations. In light of these challenges, it became evident that to incorporate as many of the stakeholders' needs and requirements, a next step was to allow experts to self-report their key figures based on a combination of their own specific data and broadly used data such as incidences.

## 1.3 Objective of this study

In this case report, we present a flexible approach that aims to examine the extent to which a given system architecture can be adapted to enable self-service reporting of required data for hospital capacity management and related researching and administration topics, such as patient disposition, gene sequencing of Covid mutants or the examination of wastewater samples, through visualizations and dashboards. We also discuss how a variety of stakeholders, who are experts in their field and data, can benefit from this approach, if integrated as content creators in such a system.

## 2 Methods

### 2.1 Analysis of the current state of the monolithic application

To understand the current state of the existing application developed in the DISPENSE project, and its functionalities, we first conducted an in-depth analysis of the data processing steps. The data flow and processing is sketched in Fig 1.

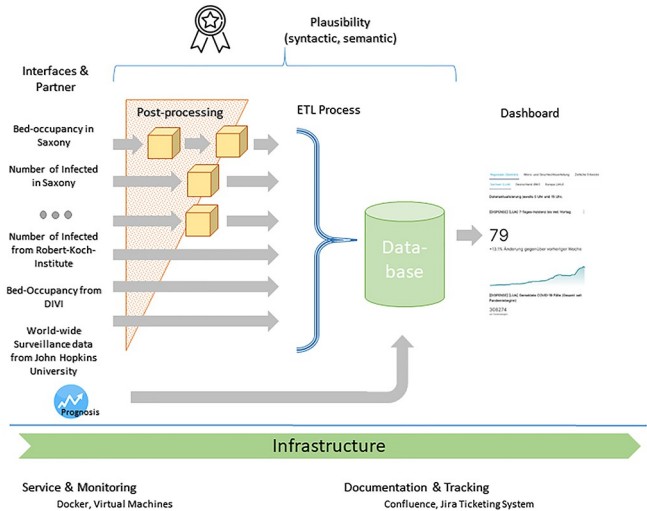

**Fig 1. Data flow of the DISPENSE tool.** Adapted from B. Lünsmann et al. [15], CC BY 4.0.

The analysis consisted of breaking down the data flow along with their individual transformation steps, its step-by-step functionalities, as well as identification of the associated user metrics.

To gain a detailed understanding of the different needs of specific user groups, such as hospital administration, the ministerial crisis management, or the patient dispatchers, we conducted a workshop. This workshop brought together the DISPENSE team who acted as facilitators who established communication with other stakeholders within the project [10] and represented the various needs of the application's user base. It's worth noting that both the members of the DISPENSE team and the authors of this paper actively play roles as stakeholders of the application, encompassing various capacities such as supplying, processing, and utilizing the data.

Participation in this project-internal workshop and the survey was voluntary among the project partners, who focused on evaluating the application and the user groups. Formal ethics approval was not required, as it included only internal stakeholders. Participants were informed orally about the voluntary nature of the study and their right to withdraw without disadvantages. Consent was obtained verbally, and the study's low-risk nature involved collecting non-sensitive feedback on personas and prototypes. Although not reviewed by an ethics committee, all procedures strictly followed ethical guidelines and adhered to the Declaration of Helsinki. All project partners consented to participate, ensuring the data collected reflected professional insights without personal or sensitive information.

For those who couldn't attend the workshop, we deployed a survey to ensure comprehensive feedback. The workshop, which is explained in detail in the Results section, aimed to identify and group the stakeholders involved in regional pandemic management in the state of Saxony. The structure of the health care system on a federal and national level, as well as all users of the application, were taken into consideration and their pandemic management tasks and interests were included.

Following a dataflow-driven approach [16, 17], which analyzes the path of data from source to target system and users, we brought together and clustered the information according to content-related service areas. In this way, we were able to understand the requirements of the different stakeholders and to identify potential areas for improvement of the platform. Using the results from the workshop, we identified functionality-related service groups.

## 2.2 Modularization of the platform's main services through use of specialized applications

Having gained an understanding of the type of reporting needed for each user group along with the associated data flows and processes, we established a concept for the system architecture to be constructed.

While methodological concepts of architectures like microservices [18, 19], offer a compelling solution to restructuring applications, transitioning to such models is often a complex and time-consuming process, usually necessitating a complete rewrite of monolithic applications. Recognizing the challenges and resource-intensive nature of this transition, we opted for a streamlined approach inspired by microservices' principles but tailored to our needs. Instead of recreating the entire system, we adopted a simplified approach to modularization and application selection and composition. This approach aimed to focus on the user requirement of enabling self-service reporting and its associated data flows, breaking down the larger tasks into main service groups. From the primary service groups of the current application, we selected specialized software tools, each tailored to handle specific tasks within the data pipeline, to manage the data flow efficiently. These tools were chosen based on their individual

capabilities that align with the unique requirements of each service group identified in Section 2.1. For example, we selected specific tools for tasks such as storing data, orchestrating tasks, visualizing data, and other functions, ensuring that each tool was best suited for its respective role in the data process.

After having modularized the application into specialized components, we integrated the previously-selected tools into a test system. This integration required individual functionalities that would allow the new applications to communicate with each other. After configuring and composing these applications accordingly, we adapted and revised the functionalities, data flows, and visualizations of the previous application. We also transferred the previous reports for pandemic management to the new system and created user groups as defined within the workshop along with their corresponding authorizations. In addition, we had to make some adjustments and develop features and extensions to cover all previous functionalities. Once these steps were completed, the previous monolithic system with its functionalities was replaced by the new, modular architecture.

## 2.3 Setting up self-service reporting

Separating the main services into individual applications also made it possible to efficiently use the visualization and reporting function with minor programming effort. In the final step, we examined whether expert groups, for example from microbiologists, physicians, statisticians, administrative staff, can customize the displayed information and reports for their specific needs independently, and identified which process steps require additional support and what capabilities are needed for self-service reporting to be possible. This helped us to understand how to best enable self-service reporting for the different user groups, and to identify any areas where additional support may be needed. As a result we created a summary of the overall design process.

## 2.4 Summary of the overall design process

The overall process for creating a self-service reporting system for regional pandemic management can be summarized as follows:

1. Analysis of the current state of the existing monolithic application

2. Workshop with stakeholders to identify and group users and their specific requirements

3. Adoption of a dataflow-driven approach to analyze data paths and create a more flexible system architecture

4. Implementation of a self-service reporting system using open-source technologies

5. Integration of the self-service reporting system into the existing pandemic management processes

6. Evaluation of the impact of the self-service reporting system on stakeholders and pandemic management.

## 3 Results

## 3.1 Identification of requirements

The workshop and survey were conducted with a total of 13 developers and user representatives from the previous monolithic application. Through this workshop, we learned that the monolithic application was primarily developed for administration and hospital management

such as the clinical coordination centers, but there was also interest from physicians, particularly for dispatching by control centers. Furthermore the workshop participants expressed strong interest in self-service reporting and data visualizations to aid them in their daily work.

With a focus on hospital administration and pandemic management, the stakeholders can be divided into three non-distinct user groups based on their role in the data flow for capacity management:

- **Data suppliers**, in particular the clinics, the health offices and the National Institute of Health and Veterinary Inspection, as well as publicly available data sources from the Robert Koch Institute, Johns Hopkins University or DIVI-Intensive Care Registry

- **Data processors** from local research facilities, such as the Center for Evidence-Based Health Care and the Center for Medical Informatics of the TU-Dresden, and

- **Data users**, including the clinical control centers for patient disposition, clinical management and legal governmental and non-governmental entities like the Saxon State Ministry for Social Affairs and Social Cohesion.

With the requirements of the different user groups, the functionalities of the application were divided into different functionality-related service groups, as shown in Table 1.

Reporting and dashboarding, as well as the user management and data processing behind them, can be divided into two subgroups: *Developers*, who use the data to create visualizations, dashboards, and reports, and *Data Viewers*, who can view the generated content, including the underlying data.

Some functionalities of the service groups overlap or build upon each other, so a distinct separation of the services is not always possible. For example, access control is necessary for managing data processing and enabling user-specific access to the data storage, but it is also necessary for the targeted release of content in the reporting or dashboarding service group, so it must be compatible with these services. Additionally, predictions themselves generate data that needs to be stored.

Since a large portion of the stakeholders are data users and focus on data visualization and reporting, we selected applications bottom-up along the data flow with the requirement of independent creation of visualizations in self-service by selected developers with expert

**Table 1. Service groups and functionalities in the DISPENSE-project.**

| Service group | Functionalities |
| --- | --- |
| Data processing approaches (in order ETL or ELT) | Extract data from source systems or APIs, |
| | Transform the datasets, and |
| | Load them into storage / target system |
| Data storage | Store the raw data, use transformation mechanisms to create user group specific datasets, and provide access to the data for service groups based on them |
| Predictions | Create forecasting models and predictions of key figures based on these models for the purpose of situation assessment |
| Content visualization | Use various statistical visualization elements, such as bar charts or line graphs, as well as key figures and also textual descriptions to provide insights into the data |
| Provide reports | Automatically generate and send reports containing the most important data and key figures for the target group, as well as descriptive texts |
| User management | Enable role-based access to data, visualizations, and reports. Specific user groups are needed to create forecasting, viewing data, evaluating data, create or view reporting |

**Table 2. Applications and descriptions for each service group.**

| Service group | Application | Description |
| --- | --- | --- |
| Data processing approaches | Apache Airflow | Orchestration tool, which brings many interfaces and processing functionalities and can handle containerized processes, as well as scripts as tasks and workflows. It was used for ELT as well as for updating model forecasts |
| Data storage | PostgreSQL-Database | Relational database for staging, data transformation and modeling, and role-based data access |
| Predictions | Custom Python and R scripts | Custom scripts for building and training predictive models and estimating forecasts from new input parameters. |
| Content visualization and provide reports | Apache superset | Application for creating charts and dashboards for data sets from data stores, enabling data exploration through role-based access control of user groups, scheduling reports and alerts. The no-code interface enables self-service development of custom metrics and visualizations. |
| User management | Keycloak | Central instance of user administration and role and access rights assignment. |

knowledge of their data. This allows us to better meet the needs of these stakeholders and enable more efficient data management and reporting.

With the requirement of using open-source applications given by the DISPENSE project, the following applications were selected exemplarily (Table 2).

## 3.2 Development of the modularized implementation

In order to deploy data flows, necessary functionalities and evaluations, each application had to be integrated and configured. This included customizing the interaction between the individual applications, which was not possible by default. We developed custom extensions for Keycloak to allow it to be used for central user management. Additionally, the few default roles in Superset were insufficient to distinguish between data viewers and developers of self-service reportings, so we established new roles. We also had to control access to specific views on a dataset level, to ensure that only authorized roles could view them. This access management in Superset had to be enabled in interaction with the organization of the preparation and schema access on the underlying PostgreSQL database, which contained the data to be visualized.

The dataflow can be described and processed through the steps displayed in Fig 2. This process enabled us to adapt the content functionalities of the previous application and provides users with access to selected data and processes after an overarching authentication. Since

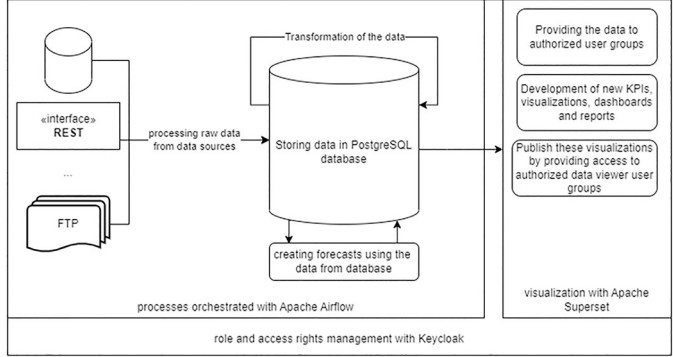

**Fig 2. The raw data was processed through Apache Airflow from data source interfaces, then migrated to a PostgreSQL database for transformation and enrichment into various target structures.** Forecasts for bed occupancies were generated from the processed data. Apache Superset was used to create new KPIs, visualizations, dashboards and standard as well as ad-hoc reports, which were made available to authorized user groups.

Superset's reporting did not support all user requirements, such as custom PDF layout, we developed an extension for Superset that generated individual LaTeX documents with Superset visualizations as PDFs, and orchestrated it using Apache Airflow. With this extension, we were able to fully adapt the existing system and functionalities to the new architecture [20]. By utilizing an application for data storage, we were able to collect and analyze retrospective data.

### 3.3 Navigating modularization during peak pandemic challenges

Initiated in winter 2020, the new system's development was a direct response to the escalating infection rates and the growing limitations of the existing application. We gave priority to critical data views, successfully delivering the initial tailored views to stakeholders by December 2020. To ensure uninterrupted data flow, the original and new systems operated concurrently. As the system evolved, additional views, including those of lesser immediacy or newly emerging data, were methodically incorporated based on their significance. By mid-2022, after comprehensive archiving, the original Shiny application was officially phased out.

### 3.4 Examples of created applications by user groups

By using dedicated applications for specific tasks, we were able to release some steps, such as the development of new visualizations, dashboards, and reports, as well as their publication, to new user groups, known as self-service developers. These are stakeholders in the data user group, who are experts in their field and can visualize their data independently in charts and compile them into dashboards under their own responsibility, which was previously done solely by the DISPENSE developers. To enable self-service developers to do this, we provided an inhouse training in data processing and visualization using Apache Superset, depending on their experience and knowledge.

For these processes, the data processors were responsible for the automated collection and preparation of the data up to the release of the data sources for the self-service developers. Developers responsible for the prediction modules were empowered to implement and orchestrate their modeling scripts independently using Apache Airflow and to interact with the database self-paced after an introduction.

Example charts and a dashboard for microbiology data can be obtained in Figs 3 and 4. The final application has currently 177 users.

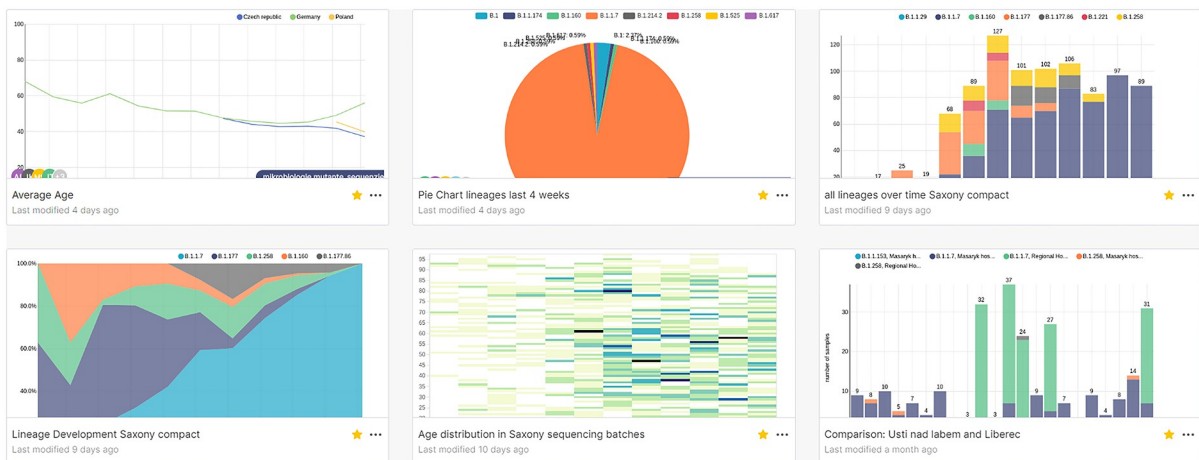

**Fig 3. Overview of various charts showing the distribution of mutants in Saxony.** This dashboard was self-made by the Institute for Medical Microbiology and Virology (TU Dresden).

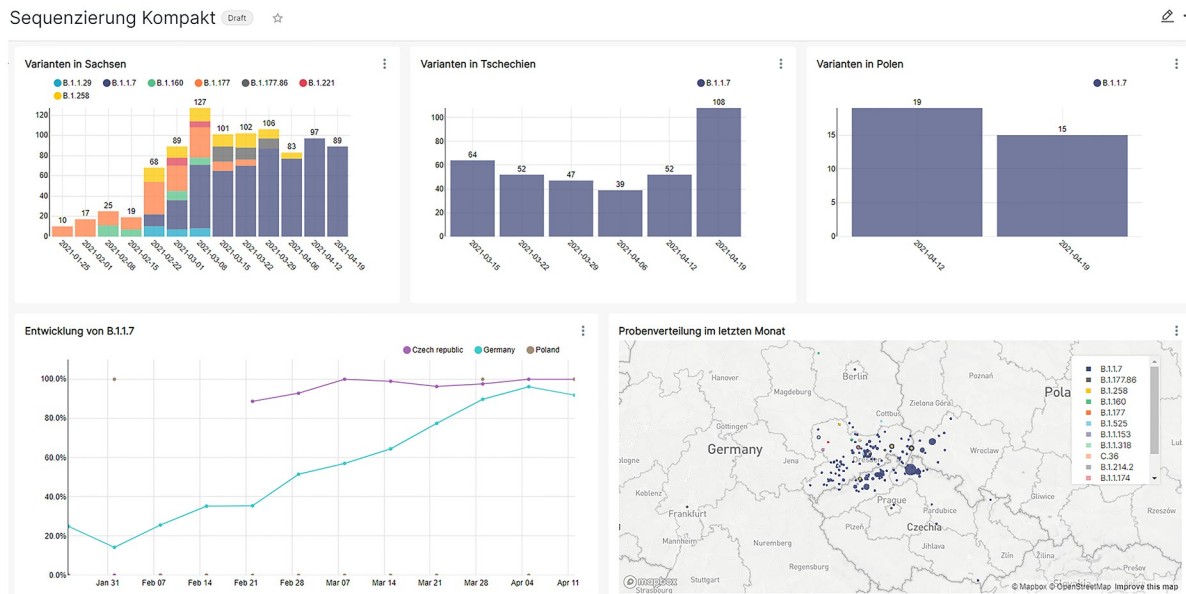

**Fig 4. Dashboard example for the distribution of virus mutants at the border regions of Saxony, which was self created by the Institute for Medical Microbiology and Virology (TU Dresden).** Map from © OpenStreetMaps, licensed under Open Data Commons Open Database License (ODbL).

## 4 Discussion

By breaking down the monolithic application into smaller, more flexible modules, we were able to address the challenges presented by rapidly changing situations, growing information needs, and associated requirements [21]. The newly implemented system allowed us to present new metrics without altering the entire data flow process and subsequent applications. We also benefited from the raw data storage feature, which enabled us to retrospectively calculate and compare metrics. Transformative tasks like data harmonization were performed on top of the raw data within the storage, making it easier to track changes and preventing processing problems or data loss.

It furthermore enabled the development of different visualizations of the data for different user groups, which could be done independently on top of either their own datasets or by granting access to certain datasets to multiple user groups. This allowed us to better meet the needs of the various user groups and provided them with the flexibility to develop and customize their own visualizations and reports based on their specific needs and interests.

Another benefit was the ability to address the high administrative efforts required to integrate new users and user groups into the applications with changing requirements for different functionalities. By implementing the interaction between user management and the individual applications, as well as enabling role-based access control, we were able to streamline this process and make it more efficient. This helped to reduce the burden on administrators and allowed them to focus on other tasks.

However, some of the individual applications, such as PostgreSQL and Apache Airflow, require specialized knowledge and programming skills, making it difficult to fully enable self-service in all service groups. As a result, only certain applications within the service groups were made available for self-service (content visualization and provide reports, user management, predictions).

Despite these limitations, the modularization of the system improved the system performance measured in an user test and increased the self-reported user-friendliness of the involved data user groups, reducing the high effort for developers required for adjustments associated with the monolithic system architecture. The user test was conducted on a selected group of users from various stakeholder groups, in particular the developers of new charts and dashboards in self-service. Some were familiar with the development of views in the previous application. They were asked to create a needed view of data themselves and then provide feedback on functionality, handling, ease of use and implementation time, and areas for improvement. While this test provided valuable insights, it was admittedly limited in scope and depth and served primarily as a checkpoint rather than a comprehensive evaluation. By enabling self-service for certain functionalities, especially the visualization and reporting services, authorized user groups were able to import new data, transform and create metrics on top of it, visualize it as charts and dashboards, and create new reports of their own, with associated advantages [22, 23]. In addition, an anonymous survey was created to evaluate users in the data user group who were only able to view a selected dashboard in Apache Superset. This involved feedback on various elements of the displays, the usability of the application and the dashboard itself, as well as communication with the developers and potential problems and requests.

Using agile software development methodologies the development process was both iterative and responsive to the dynamic needs of the pandemic. Based on the SCRUM framework, a term derived from rugby, representing close team collaboration and concentrated phase-based work, we worked in two-week sprint cycles to ensure rapid development and deployment of new features. Within each sprint, we used Kanban to coordinate our tasks. This allowed for rapid adaptation to shifting stakeholder requirements. Even as we progressed with the new system, the original application remained functional, ensuring no interruption in the vital flow of data. Additionally, the versatility of the modularized system is evident, not just in its relevance to COVID-19 but in its potential applicability to other healthcare scenarios. For instance, it could have been instrumental in analyzing the RS virus impact of 2022 in Germany or the ramifications of events such as the Ukraine crisis or heat waves on hospital capacities.

The specialization of single applications on single service groups allowed users to focus on content visualization and reporting without needing knowledge in other areas, such as programming data processing tasks. This helped to make the system more scalable and user-friendly.

## 5 Conclusion

In conclusion, the modularization of the monolithic COVID-19 pandemic management application in the federal State of Saxony, Germany, was successful in addressing the challenges of constantly changing information needs, frequent adjustments to the (source) systems, and the need for self-service reporting. By dividing the main services into specialized applications, it was possible in our work to reduce the administrative effort required for integrating new users and user groups, as well as enabling role-based access control. Based on specific user feedback, the system also became more efficient, as the system's availability was increased and outages were reduced, and user-friendly, as users were able to focus on content visualization and reporting without needing knowledge in other areas, like programming. However, there were also limitations to this approach, including the complexity of the individual applications and the associated high training costs, as well as the lack of self-service data engineering features in some applications. Despite these limitations, the modularization of the system was a valuable step towards improving the collaboration between research, care, and management in Saxony

and beyond during the COVID-19 pandemic. It may serve as a template for upcoming or existing studies that may need a transfer from a monolithic to a modularized architecture.

## Acknowledgments

We thank our colleagues of the University Hospital in Dresden and the TU Dresden for providing the technical infrastructure for this project, in particular, Stephan Lorenz, Christoph Forkert, Felix Walther, Fabian Baum, Andreas Mogwitz, Robin R. Weidemann, Christian Kleber and Kathleen Juncken as well as our colleagues in Chemnitz and Leipzig, especially Nicole Lakowa and Sebastian N. Stehr. We are also thankful for the support of the State Health and Veterinary Research Institute ("Sächsische Landesuntersuchungsanstalt"), the regional health departments and Professor Röder and his staff (TU Dresden).

We would like to extend our sincere gratitude to the Institute for Medical Microbiology and Virology (TU Dresden), especially Marlena Schlecht, Alexa Laubner, and Prof. Alexander Dalpke, who is now leading the Research Group Dalpke at the Heidelberg University Hospital. Not only did they provide invaluable examples of the self-service functionality that enriched our presentation, but their consistent feedback as users was instrumental in refining and enhancing the software's features. Their dedication and engagement have significantly contributed to the software's effectiveness and user-friendliness.

## Author Contributions

**Conceptualization:** Richard Gebler, Martin Lehmann, Markus Wolfien, Veronika Bierbaum, Toni Lange, Jochen Schmitt, Martin Sedlmayr.

**Data curation:** Richard Gebler, Martin Lehmann, Mirko Gruhl, Veronika Bierbaum, Toni Lange, Katja Polotzek, Hanns-Christoph Held.

**Formal analysis:** Richard Gebler, Maik Löwe, Veronika Bierbaum, Katja Polotzek.

**Funding acquisition:** Hanns-Christoph Held, Michael Albrecht, Jochen Schmitt, Martin Sedlmayr.

**Investigation:** Richard Gebler, Maik Löwe, Veronika Bierbaum.

**Methodology:** Richard Gebler, Martin Lehmann, Maik Löwe, Veronika Bierbaum.

**Project administration:** Richard Gebler, Veronika Bierbaum, Hanns-Christoph Held, Michael Albrecht, Jochen Schmitt, Martin Sedlmayr.

**Resources:** Richard Gebler, Martin Lehmann, Maik Löwe, Mirko Gruhl, Veronika Bierbaum, Katja Polotzek, Hanns-Christoph Held, Michael Albrecht, Jochen Schmitt, Martin Sedlmayr.

**Software:** Richard Gebler, Martin Lehmann, Maik Löwe, Mirko Gruhl, Andreas Hasselberg, Veronika Bierbaum, Toni Lange.

**Supervision:** Richard Gebler, Maik Löwe, Veronika Bierbaum, Jochen Schmitt, Martin Sedlmayr.

**Validation:** Richard Gebler, Mirko Gruhl, Markus Wolfien, Veronika Bierbaum, Jochen Schmitt, Martin Sedlmayr.

**Visualization:** Richard Gebler, Martin Lehmann, Mirko Gruhl, Veronika Bierbaum, Katja Polotzek.

**Writing – original draft:** Richard Gebler, Markus Wolfien, Veronika Bierbaum.

**Writing – review & editing:** Richard Gebler, Martin Lehmann, Mirko Gruhl, Markus Wol-
fien, Miriam Goldammer, Franziska Bathelt, Jens Karschau, Veronika Bierbaum, Jochen
Schmitt, Martin Sedlmayr.

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
