## [Decision Letter · Decision Letter 0]

19 Sep 2023

PONE-D-23-22811Supporting regional pandemic management by enabling self service reporting through modularisationPLOS ONE

Dear Dr. Gebler,

Thank you for submitting your manuscript to PLOS ONE. After careful consideration, we feel that it has merit but does not fully meet PLOS ONE’s publication criteria as it currently stands. Therefore, we invite you to submit a revised version of the manuscript that addresses the points raised during the review process.

We look forward to receiving your revised manuscript.

Kind regards,

Anna Bernasconi, PhD

Academic Editor

PLOS ONE

“Outside the scope of this study, Jochen Schmitt has received consultation fees from Novartis, Lilly, and Sanofi, and his institution has been granted funding for investigator-initiated research from Novartis, Sanofi, Pfizer, and ALK. Michael Albrecht holds the position of CEO at the University Hospitals Carl Gustav Carus Dresden. We wish to clarify that these affiliations do not influence our commitment to PLOS ONE's policies on data and material sharing.”

5. Please amend either the title on the online submission form (via Edit Submission) or the title in the manuscript so that they are identical.

6. Please remove your figures from within your manuscript file, leaving only the individual TIFF/EPS image files, uploaded separately. These will be automatically included in the reviewers’ PDF.

7. We note that Figures 1, 2 and 3 in your submission contain copyrighted images. All PLOS content is published under the Creative Commons Attribution License (CC BY 4.0), which means that the manuscript, images, and Supporting Information files will be freely available online, and any third party is permitted to access, download, copy, distribute, and use these materials in any way, even commercially, with proper attribution. For more information, see our copyright guidelines: http://journals.plos.org/plosone/s/licenses-and-copyright.

1. You may seek permission from the original copyright holder of Figures 1, 2 and 3 to publish the content specifically under the CC BY 4.0 license.

8. We note that Figure 1 in your submission contain [map/satellite] images which may be copyrighted. All PLOS content is published under the Creative Commons Attribution License (CC BY 4.0), which means that the manuscript, images, and Supporting Information files will be freely available online, and any third party is permitted to access, download, copy, distribute, and use these materials in any way, even commercially, with proper attribution. For these reasons, we cannot publish previously copyrighted maps or satellite images created using proprietary data, such as Google software (Google Maps, Street View, and Earth). For more information, see our copyright guidelines: http://journals.plos.org/plosone/s/licenses-and-copyright.

Additional Editor Comments:

Dear authors, the reviewers have appreciated the proposed work. Please revise it taking into account their comments.

Reviewers' comments:

Reviewer's Responses to Questions

**Comments to the Author**

1. Is the manuscript technically sound, and do the data support the conclusions?

Reviewer #1: Yes

Reviewer #2: Yes

2. Has the statistical analysis been performed appropriately and rigorously? 

Reviewer #1: N/A

Reviewer #2: N/A

3. Have the authors made all data underlying the findings in their manuscript fully available?

Reviewer #1: Yes

Reviewer #2: Yes

4. Is the manuscript presented in an intelligible fashion and written in standard English?

Reviewer #1: Yes

Reviewer #2: Yes

5. Review Comments to the Author

Reviewer #1: Thank you for the opportunity to review this article. It was a clearly written and interesting paper though requires clarification of some points to increase understanding of the context of the paper.

I offer the following comments to the authors:

• There appears to be some citations missing from the introductory section where there are two references to citations in brackets but no author/paper reference.

• In the introduction the authors refer to the use of the data base as an aid to allocating health care resources. However, it is not clear (at this point) what health care resources are being addressed and it would bear some explanation. For example, are the resources referred to equipment, bed spaces, personnel, or other. Later in the paper, it appears that the main focus of the data is volume of cases and bed management, but it would be helpful if this was explicit at an earlier point.

• Similarly, some context in relation to the team that conducted the study and developed the application and their relationship to stakeholders was unclear. The paper discussed the need to bring research, data, and practice together, but it is unclear if the authors of the paper were data base developers, analysts, researchers, or had another relationship to the stakeholders. It would be helpful to understand more about the context for their involvement. Although we are given some information about the DISPENSE project, there is no clear explanation of how this operates, where it is based, how it is led and funded, and what the relationship is between the paper authors and the project. It would also be helpful to clarify earlier in the paper, the nature of the stakeholder/user groups and their level of seniority and influence in their respective organisations in relation to the use of the database application.

• It is not clear if the updates and upgrades to the functionality impacted on the value to the users throughout the pandemic or whether the upgrades interrupted the flow of data. How agile did the system prove to be? At what stage in the pandemic did a functional system become available and when did the data flow start to support the management of the crisis?

• According to the paper the data application has been in development since March 2020. Are we to understand that the modifications were completed in a specific time scale between 2020 and 2023 or are still under review. It would be helpful to clarify the time scale for the project and whether the model is still being used in the context of Coronavirus or in a different context.

• In the discussion the authors refer to a number of benefits to the users of the application. However, there is no indication of a mechanism for user feedback or of an evaluation having taken place to ascertain the perspective of the users. A workshop was held to identify user needs and stage 6 of the process for the self-reporting system is ‘evaluation of the impact’ but it is not clear if there was any follow up to confirm the users’ needs were met. The discussion suggests that the impact and measure of success is taken from the perspective of the reporting team. A ‘user test’ is mentioned but not reported in detail. It would be helpful if the authors could clarify if an evaluation has taken place.

I hope these comments are helpful to the authors in revising their paper.

Reviewer #2: The DISPENSE project was created to predict short-term hospital bed capacity demands in Saxony, Germany. In this case report, the authors conducted an in-depth analysis of the data processing steps. They interviewed the project's developers for

The results propose a set of applications that could be implemented by other teams for the same type of project.

Major comments :

The introduction describes at length how the application is set up and how it works. I propose to move the technical details of this implementation to a Material sub-section of the Methods section. Furthermore, in the reference, no reference is made to equivalent initiatives to adapt existing systems (such as, for exemple, the reference 15).

Minor comments :

Two citations are missing page 2, lines 59 and 62.

Page 7 lines 160-162 : Did the stakeholders take part in the workshop? It is not comprehensive.

Page 7 line 180 : Does it means specialized software for the ETL ?

Page 8 line 206 : "To understand the initial situation of the application and user groups, we conducted an internal workshop, and a survey for those who could not attend". This seems more appropriate to a method section.

Page 15 Line 303 to 317 : this list is what is missing in the method, in order to have a complete overview of the process. Why is it the last part of the results ?

6. PLOS authors have the option to publish the peer review history of their article (what does this mean?). If published, this will include your full peer review and any attached files.

Reviewer #1: **Yes: **Dr Virginia Minogue

Reviewer #2: **Yes: **Antoine Lamer

---

## [Author Response · Author response to Decision Letter 0]

21 Nov 2023

Reviewer 1:

Thank you for the opportunity to review this article. It was a clearly written and interesting paper though requires clarification of some points to increase understanding of the context of the paper.

I offer the following comments to the authors:

Comment: “There appears to be some citations missing from the introductory section where there are two references to citations in brackets but no author/paper reference.”

Answer: Thank you for pointing out the missing of citations in the introductory section. An oversight occurred during the manuscript preparation, and these citations were accidentally left out in the final version submitted. We have now corrected this mistake. The two missing citations have been added as references 6 and 7. Furthermore, two follow-up citations (8,9) were not linked properly, which is now corrected. Subsequent references have been renumbered accordingly. We appreciate your thorough review and attention to detail, and apologize for any confusion this may have caused.

Comment: “In the introduction the authors refer to the use of the data base as an aid to allocating health care resources. However, it is not clear (at this point) what health care resources are being addressed and it would bear some explanation. For example, are the resources referred to equipment, bed spaces, personnel, or other. Later in the paper, it appears that the main focus of the data is volume of cases and bed management, but it would be helpful if this was explicit at an earlier point.”

Answer: Thank you for your constructive feedback on clarifying the specific nature of "healthcare resources" in the introduction. Based on your recommendation, we have revised the relevant section of the introduction to emphasize that a primary focus of the data and the subsequent resource allocation was on bed management in hospitals. The revised sentences are as follows:

"As soon as local outbreaks emerged, many occurred at healthcare facilities like nursing homes, affecting particularly vulnerable groups. This situation heightened the demand for more granular regional information. A primary focus became bed management in hospitals, as there was a need to effectively allocate healthcare resources to areas with high infection rates."

We believe this revision provides clearer insight early in the paper about the main resource in focus. We appreciate your keen observation and hope this change addresses your concerns.

Comment: “Similarly, some context in relation to the team that conducted the study and developed the application and their relationship to stakeholders was unclear. The paper discussed the need to bring research, data, and practice together, but it is unclear if the authors of the paper were data base developers, analysts, researchers, or had another relationship to the stakeholders. It would be helpful to understand more about the context for their involvement. Although we are given some information about the DISPENSE project, there is no clear explanation of how this operates, where it is based, how it is led and funded, and what the relationship is between the paper authors and the project. It would also be helpful to clarify earlier in the paper, the nature of the stakeholder/user groups and their level of seniority and influence in their respective organisations in relation to the use of the database application.”

Answer: Thank you for your invaluable feedback highlighting the need for clarity regarding the roles and relationship of the authors to the DISPENSE project and its stakeholders. In response to your comments, we have made modifications to provide greater clarity and context. In particular, we have included the following statements:

The DISPENSE project was “[...]funded and sanctioned by the Saxon Ministry for Social Affairs[...]” to provide a clear understanding of the funding and overarching support structure for the project.

“The authors of this paper were deeply involved in the development and operational aspects of this project, working alongside other professionals to ensure its effectiveness.” This statement has been added to clarify the active and multifaceted roles the authors played within the DISPENSE project.

“Given the authors' involvement in this project, which ranged from data collection, preparation, and forecasting to system development and dashboarding, the insights shared here come from their first-hand experience.” This serves to underline the hands-on experience and expertise the authors bring to this case report, covering the entire spectrum of the project's functionalities.

We hope that with these additions, the nature of the authors' relationship to the DISPENSE project and its stakeholders is now more evident. We appreciate your thorough review, which has helped in refining the manuscript for better clarity and comprehension.

Comment: It is not clear if the updates and upgrades to the functionality impacted on the value to the users throughout the pandemic or whether the upgrades interrupted the flow of data.

How agile did the system prove to be? 

At what stage in the pandemic did a functional system become available and 

when did the data flow start to support the management of the crisis?

And “According to the paper the data application has been in development since March 2020. Are we to understand that the modifications were completed in a specific time scale between 2020 and 2023 or are still under review. It would be helpful to 

clarify the time scale for the project and 

whether the model is still being used in the context of Coronavirus or in a different context.”

Answer: Thank you for your thoughtful questions and for pointing out areas that required more clarity. In response to your feedback, we have added details to our manuscript to elucidate the evolution of the system, its agility, its deployment timeline, and its broader applicability.

1. System agility & impact on value:

Using agile software development methodologies ensured that our process was both iterative and responsive to the dynamic needs of the pandemic. 

Therefore, we added the following text to the discussion section:

“Using agile methods and the SCRUM framework, our development was iterative and responded immediately to the changing needs of the pandemic. We worked in two-week sprints, with Kanban guiding task coordination and ensuring rapid adaptation to stakeholders' needs. At the same time, the original application remained active to ensure a continuous flow of data.”

 We made it a priority to keep the original application functional throughout, thus ensuring there was no interruption in the crucial data flow to the stakeholders.

2. System deployment timeline:

In our new section titled "Navigating modularization during peak pandemic challenges", we detail the timeline of the system's development:

“Initiated in winter 2020, the new system's development was a direct response to the escalating infection rates and the growing limitations of the existing application. We gave priority to critical data views, successfully delivering the initial tailored views to stakeholders by December 2020. To ensure uninterrupted data flow, the original and new systems operated concurrently. As the system evolved, additional views, including those of lesser immediacy or newly emerging data, were methodically incorporated based on their significance. By mid-2022, after comprehensive archiving, the original Shiny application was officially phased out.”

3. Ongoing relevance of the system:

In the discussion, we have also emphasized the modularized system's versatility:

“Additionally, the versatility of the modularized system is evident, not just in its relevance to COVID-19 but in its potential applicability to other healthcare scenarios. For instance, it could have been instrumental in analyzing the RS virus impact of 2022 in Germany or the ramifications of events such as the Ukraine crisis or heat waves on hospital capacities.”

Presently, while its immediate relevance to COVID-19 remains, its modular nature means it can be repurposed and adapted for various other healthcare challenges and scenarios.

Comment: “In the discussion the authors refer to a number of benefits to the users of the application. However, there is no indication of a mechanism for user feedback or of an evaluation having taken place to ascertain the perspective of the users. A workshop was held to identify user needs and stage 6 of the process for the self-reporting system is ‘evaluation of the impact’ but it is not clear if there was any follow up to confirm the users’ needs were met. The discussion suggests that the impact and measure of success is taken from the perspective of the reporting team. A ‘user test’ is mentioned but not reported in detail. It would be helpful if the authors could clarify if an evaluation has taken place. “

Answer: Thank you for your thoughtful feedback on our discussion section. We acknowledge your concerns regarding the evaluation and user feedback on our system. To address these concerns, we clarified the process by adding the following two segments to the discussion section.

The user test was conducted on a selected group of users from various stakeholder groups, especially the developers of new charts and dashboards in self-service. The majority was familiar with the development of views in the previous application. They were asked to create a needed view of data themselves and then provide feedback on functionality, handling, ease of use and implementation time, and areas for improvement. While this test provided valuable insights, it was admittedly limited in scope and depth and served primarily as a checkpoint rather than a comprehensive evaluation.

[...]

This involved feedback on various elements of the displays, the usability of the application and the dashboard itself, as well as communication with the developers and potential problems and requests. 

Reviewer #2: The DISPENSE project was created to predict short-term hospital bed capacity demands in Saxony, Germany. In this case report, the authors conducted an in-depth analysis of the data processing steps. They interviewed the project's developers for

The results propose a set of applications that could be implemented by other teams for the same type of project.

Major comments :

The introduction describes at length how the application is set up and how it works. I propose to move the technical details of this implementation to a Material sub-section of the Methods section. 

Answer: 

Thank you for your valuable feedback on the structure of our manuscript. We truly appreciate your insights and have taken steps to address your concerns and improve the overall clarity and flow of the paper.

In light of your feedback, we have introduced specific subheadings in the introduction:

1.1 The rise of regional data management needs

1.2 The initial challenges with DISPENSE

1.3 Objective of this study

We believe that by distinguishing between the broader global context, the emergence of regional data management needs, the initial challenges with DISPENSE, and the main objective of our study, readers will be able to follow the evolution and grasp the context more easily.

Your reference to the detailed description of the initial setup of the application is very apt. In particular, in Section 1.3 The Initial Challenges with DISPENSE, we explained the initial state of the DISPENSE tool, which we consider the state of the art on which this study is built. We felt it was extremely important to provide this detailed foundation in the introduction to help readers understand the challenges we were trying to address. This context provides the foundation for the sections that follow, in which we discuss our methods, results, and proposed solutions.

The presentation of the "state of the art" in the introduction through 1.3 is essential because it highlights the existing challenges and underscores the need for the research and solutions presented in this paper. It allows the reader to understand the initial conditions and constraints before delving into our proposed solutions and the technical details in the Methods and Results sections.

We recognize the importance of a well-structured and reader-friendly paper and have strived to achieve this balance in our revisions. We are confident that the introduction of these specific subheadings and further detail on the original DISPENSE challenges will improve the clarity and flow of the manuscript.

We thank you again for your constructive feedback. We hope that our revisions are consistent with your suggestions and improve the overall quality of the paper.

Comment: “Furthermore, in the reference, no reference is made to equivalent initiatives to adapt existing systems (such as, for exemple, the reference 15).”

Answer: Thank you for pointing out the relevance of discussing comparable initiatives in the adaptation of existing systems. We value your feedback and have made revisions to provide clarity on our methodological choices. Specifically, we felt it was crucial to explain our rationale for not fully adopting architectures like microservices.

In the revised Methods section, we now state: “While methodological concepts of architectures like microservices, offer a compelling solution to restructuring applications, transitioning to such models is often a complex and time-consuming process, usually necessitating a complete rewrite of monolithic applications. Recognizing these challenges and resource-intensive nature of this transition, we opted for a streamlined approach inspired by microservices' principles but tailored to our needs. Instead of recreating the entire system, we adopted a simplified approach to modularization and application selection and composition. This approach aimed to focus on the user requirement of enabling self-service reporting and its associated data flows, breaking down the larger tasks into main service groups. From the primary service groups of the current application, we selected specialized software tools, each tailored to handle specific tasks within the data pipeline, to manage the data flow efficiently. These tools were chosen based on their individual capabilities that align with the unique requirements of each service group identified in Section 2.1. For example, we selected specific tools for tasks such as storing data, orchestrating tasks, visualizing data, and other functions, ensuring that each tool was well suited for its respective role in the data process.”

We trust that this modification offers a clearer perspective on our decision-making process and the steps taken. Your insights have significantly enriched our manuscript, and we are deeply grateful for your constructive feedback.

Minor comments :

Comment: “Two citations are missing page 2, lines 59 and 62.”

Answer: Thank you for noting the missing citations in the introduction. I made an error when finalizing the manuscript, which left out these citations. I've added the missing ones as references 6 and 7. Also, I fixed two other citations (8,9) that weren't linked right. All the other references have been adjusted because of these changes. I'm grateful for your careful review and apologize for the oversight.

Page 7 lines 160-162 : Did the stakeholders take part in the workshop? It is not comprehensive.

AND comment Page 8 line 206 : "To understand the initial situation of the application and user groups, we conducted an internal workshop, and a survey for those who could not attend". This seems more appropriate to a method section

Answer: Thank you for your comment. We have revised the section to clarify our approach and to provide a more detailed account. Now, the manuscript states:

“To gain a detailed understanding of the different needs of specific user groups, such as hospital administration, the ministerial crisis management, or the patient dispatchers, we conducted a workshop. This workshop brought together the DISPENSE team who acted as facilitators in establishing communication with other stakeholders within the project [10] and represented the various needs of the application's user base. It is worth noting that both the members of the DISPENSE team and the authors of this paper actively play roles as stakeholders of the application. This encompasses various capacities, such as supplying, processing, and utilizing the data. For those who could not attend the workshop, we deployed a survey to ensure comprehensive feedback. The workshop, which is explained in detail in the Results section, aimed to identify and group the stakeholders involved in regional pandemic management in the state of Saxony, Germany. The structure of the health care system on a federal and national level, as well as all users of the application, were taken into consideration and their pandemic management tasks and interests were included.“

The beginning of the result section part is shortened to:

“The workshop and survey were conducted with a total of 13 developers and user representatives of the previous monolithic application.”

By this, we aim to emphasize that the workshop was not an internal event but involved developers from the DISPENSE Project, thereby representing a wide spectrum of application users and stakeholder interests. We hope these revisions make our methodology and participants more transparent.

Page 7 line 180 : Does it means specialized software for the ETL ?

Answer: Thank you for your constructive feedback on clarifying the selection of software tools within our methods section. To address your concern about the specificity of the software selection, we have revised the section to better elucidate our approach. The updated passage now reads:

“From the primary service groups of the current application, we selected specialized software tools, each tailored to handle specific tasks within the data pipeline, to manage the data flow efficiently. These tools were chosen based on their individual capabilities that align with the unique requirements of each service group identified in Section 2.1. For example, we selected specific tools for tasks such as storing data, orchestrating tasks, visualizing data, and other functions, ensuring that each tool was best suited for its respective role in the data process.”

Page 15 Line 303 to 317 : this list is what is missing in the method, in order to have a complete overview of the process. Why is it the last part of the results ?

Answer: Thank you for your valuable feedback. We have taken your suggestion into account and shifted the content to the end of the methods section, specifically to the new section 2.4. We believe that detailing the complete process in the methods section will provide readers with a comprehensive understanding of the steps undertaken, thereby enabling them to replicate the process if needed. Although the process generated the positive results, the steps we took are indeed methodological in nature and fit better within the methods section.

---

## [Decision Letter · Decision Letter 1]

28 Dec 2023

Supporting regional pandemic management by enabling self-service reporting - a case report

PONE-D-23-22811R1

Dear Dr. Gebler,

We’re pleased to inform you that your manuscript has been judged scientifically suitable for publication and will be formally accepted for publication once it meets all outstanding technical requirements.

Kind regards,

Anna Bernasconi, PhD

Academic Editor

PLOS ONE

Additional Editor Comments (optional):

Dear authors, thank you for providing a revised version of your manuscript and complete rebuttal. One reviewer and myself have thoroughly evaluated this versione: the manuscript has been improved and is now of appropriate quality for the journal; please mind the last comment of Reviewer 1 when preparing the final version.

Reviewers' comments:

Reviewer's Responses to Questions

**Comments to the Author**

1. If the authors have adequately addressed your comments raised in a previous round of review and you feel that this manuscript is now acceptable for publication, you may indicate that here to bypass the “Comments to the Author” section, enter your conflict of interest statement in the “Confidential to Editor” section, and submit your "Accept" recommendation.

Reviewer #1: All comments have been addressed

2. Is the manuscript technically sound, and do the data support the conclusions?

Reviewer #1: Yes

3. Has the statistical analysis been performed appropriately and rigorously? 

Reviewer #1: Yes

4. Have the authors made all data underlying the findings in their manuscript fully available?

Reviewer #1: No

5. Is the manuscript presented in an intelligible fashion and written in standard English?

Reviewer #1: Yes

6. Review Comments to the Author

Reviewer #1: I am content the authors have addressed the issues outlined in my review and thank them for their full response to the review. There is one further, minor, issue I suggest needs addressing. The acronym SCRUM is used in relation to a framework they used. The acronym is not explained nor is it referenced. This should be resolved prior to publication.

7. PLOS authors have the option to publish the peer review history of their article (what does this mean?). If published, this will include your full peer review and any attached files.

Reviewer #1: **Yes: **Dr Virginia Minogue

---

## [Editor Report · Acceptance letter]

22 Jan 2024

PONE-D-23-22811R1 

PLOS ONE

Dear Dr. Gebler, 

I'm pleased to inform you that your manuscript has been deemed suitable for publication in PLOS ONE. Congratulations! Your manuscript is now being handed over to our production team.

Kind regards, 

on behalf of

Dr. Anna Bernasconi 

Academic Editor

PLOS ONE